# Smart Graphene-Based Electrochemical Nanobiosensor for Clinical Diagnosis: Review

**DOI:** 10.3390/s23042240

**Published:** 2023-02-16

**Authors:** Irkham Irkham, Abdullahi Umar Ibrahim, Pwadubashiyi Coston Pwavodi, Fadi Al-Turjman, Yeni Wahyuni Hartati

**Affiliations:** 1Department of Chemistry, Faculty of Mathematics and Natural Sciences, Padjadjaran University, Bandung 40173, Indonesia; 2Department of Biomedical Engineering, Near East University, Mersin 10, Nicosia 99010, Turkey; 3Department of Bioengineering/Biomedical Engineering, Faculty of Engineering, Cyprus International University, Haspolat, North Cyprus via Mersin 10, Nicosia 99010, Turkey; 4Research Center for AI and IoT, Faculty of Engineering, University of Kyrenia, Mersin 10, Kyrenia 99320, Turkey; 5Artificial Intelligence Engineering Department, AI and Robotics Institute, Near East University, Mersin 10, Nicosia 99010, Turkey

**Keywords:** biosensors, graphene, electrochemical sensors, nanocomposites, artificial intelligence, Internet of Medical Things (IoMT), nanoparticles

## Abstract

**Highlights:**

**What are the main findings?**
Point-of-care diagnosis is crucial for management of infectious diseases.Integration of nanotechnology into biosensing technology increases conductivity, sensitivity and Limit of Detection (LOD).

**What is the implication of the main finding?**
Graphene-based electrochemical biosensors have emerged as one of the best approaches for enhancing biosensing technology.Integration of Internet of Medical Things (IoMT) in the development of biosensors have the potential to improve detection of diseases and treatments.

**Abstract:**

The technological improvement in the field of physics, chemistry, electronics, nanotechnology, biology, and molecular biology has contributed to the development of various electrochemical biosensors with a broad range of applications in healthcare settings, food control and monitoring, and environmental monitoring. In the past, conventional biosensors that have employed bioreceptors, such as enzymes, antibodies, Nucleic Acid (NA), etc., and used different transduction methods such as optical, thermal, electrochemical, electrical and magnetic detection, have been developed. Yet, with all the progresses made so far, these biosensors are clouded with many challenges, such as interference with undesirable compound, low sensitivity, specificity, selectivity, and longer processing time. In order to address these challenges, there is high need for developing novel, fast, highly sensitive biosensors with high accuracy and specificity. Scientists explore these gaps by incorporating nanoparticles (NPs) and nanocomposites (NCs) to enhance the desired properties. Graphene nanostructures have emerged as one of the ideal materials for biosensing technology due to their excellent dispersity, ease of functionalization, physiochemical properties, optical properties, good electrical conductivity, etc. The Integration of the Internet of Medical Things (IoMT) in the development of biosensors has the potential to improve diagnosis and treatment of diseases through early diagnosis and on time monitoring. The outcome of this comprehensive review will be useful to understand the significant role of graphene-based electrochemical biosensor integrated with Artificial Intelligence AI and IoMT for clinical diagnostics. The review is further extended to cover open research issues and future aspects of biosensing technology for diagnosis and management of clinical diseases and performance evaluation based on Linear Range (LR) and Limit of Detection (LOD) within the ranges of Micromolar µM (10^−6^), Nanomolar nM (10^−9^), Picomolar pM (10^−12^), femtomolar fM (10^−15^), and attomolar aM (10^−18^).

## 1. Introduction

Diagnosis of a disease is one of the most important aspects of clinical medicine. Medical experts such as clinicians and medical laboratory technologies use biological samples which include nucleic acids, proteins, microorganisms, and other related biological molecules to diagnose patients suffering from various diseases. Thus, detection of disease-causing agents and markers is one of the most crucial and fundamental parts of human health diagnostics and treatment. One of the challenges faced by healthcare experts in terms of diagnostics of diseases includes low sensitivity and specificity, longer diagnosis time, high cost, interference with undesirable compound, false positive result, the use of toxic chemicals, etc. This calls for the need to develop non-invasive, specific, sensitive, rapid, and cheap detection assays [1,2].

As a result of transformation in science and technology, scientists over the years have developed several biosensors for clinical diagnostics applications such as detection of glucose level, the presence of pathogenic-causing microbes, cancer markers, protein markers, and hematological malignances. In order to improve the required pM limit of detection (LOD) of biological samples, scientists incorporated nanomaterials for the coatings of transducers for easy conversion and for enhancing bioreceptor immobilization in order to improve signal readout [3].

In order to solve some of these challenges, scientists integrated several nanostructure materials. Nanoparticles have been extensively integrated in a wide range of biomedical applications such as biochemical diagnosis, imaging, drug delivery, and treatment due to their unique properties, which include large surface area, small size, and physical and chemical properties. Their bioanalytical applications in terms of modification of electrodes has made them suitable for improving the sensitivity and selectivity of biomedical sensors [4]. Several nanostructured materials have been employed in biosensing which include metallic NPs such as gold, platinum, aluminum, iron, Cobalt, palladium, copper, metallic oxides such as titanium oxides, zinc oxides, manganese oxides, and other nanostructured materials such as quantum dots, carbon-based nanomaterials, dendric fibers, silicon fibers, etc., [5].

The incorporation of nanostructured materials has been shown to improve several characteristics of biomedical sensors in terms of selectivity, conductivity, sensitivity, etc. However, despite the progress made in the last five decades, clinical biosensors are still hindered by many challenges which include the need for point-of-care (POC) diagnostics, picomolar (pM) sensitivity, fast response time, and high specificity [3]. Thus, there is high need to develop biosensors with desirable characteristics by incorporating nanomaterials or nanocomposites with outstanding electrical, optical, mechanical, and chemical properties [6].

The integration of graphene in biosensing technology has shown high potential in solving limitations that clouded the use of biosensors for POC diagnosis. Graphene possesses many desired properties which makes it an ideal component for biosensing applications. Some of the advantages of graphene-based biosensors over other types of biosensors include a small band gap which is useful for sensitive electrochemical and electrical readout, excellent conductivity, high surface to volume ratio, and optical properties (such as plasmonic and fluorescence) for optical readouts [7,8].

The different forms of nanographene considered include graphene nanoribbons, graphene oxide (GO), and reduced graphene oxide (rGO). The integration of Artificial Intelligence (AI) in biosensing technology has played a vital role in data processing, signal acquisition and transportation, decision system, biorecognition element investigation. and material innovation [9].

Engineering of POCT can play a significant role in preventing the transmission of diseases through on-site rapid diagnosis and real time testing. One of the emerging fields that can support the development of POCT devices is Internet of Medical Things (IoMT). POCT devices can be integrated with IoMT to offer connectivity and wireless-based operation between devices and medical expert for timely and accurate decision making [10]. In this review, the potential of a graphene-based electrochemical biosensor integrated with IoMT for clinical diagnosis is discussed.

### 1.1. Comparison with Similar Studies

Scientists have contributed extensively to the field of biosensors, graphene chemistry and applications, diagnosis of diseases, and the relevancy of medical data in the 21st century healthcare setting. In this subsection, we overview existing surveys and discussed how they differ with our review.

The study provided by Fracchiolla et al. [11] offers an extensive overview of biosensors and clinical applications as well as challenges and future prospective of biosensors in clinical diagnostics. Nonetheless, the study does not cover the application of nanomaterials in biosensing technology and medical data. One of the reviews that is somehow similar to our review is the one given by Szunerits and Boukherroub [12]. The authors discussed different types of biosensors, the potential of graphene-nanomaterials, and clinical diagnostics of diseases. However, the authors have not extensively discussed POC diagnostics, medical data, and open research issues.

The review conducted by Krishnan et al. [13] discusses graphene-based nanocomposites for electrochemical and fluorescent biosensors. The authors overview 2D and 3D graphene-based nanocomposites, properties, surface functionalization, physical adsorption, enzyme inhibition strategies, etc., and how they enhanced the electrochemical and fluorescent biosensing mechanisms for detection of different biomolecules. Despite the wide range of topics covered, authors have not elaborated on POC diagnostics, acquisition of medical data, or the use of AI and IoMT.

The review provided by Jain et al. [10] focuses on the use of biosensors for clinical diagnosis of infectious diseases (emerging and re-emerging diseases which include SARS-CoV-2, Influenza virus, Zika virus, Dengue virus, Ebola virus, and Malaria) and the potential of IoMT assisted point-of-care diagnostics. Despite all the shared similarities with our review, the article does not present intensive overview of biosensors, nanomaterials, and the unique conductivity of Graphene which makes it one of the best nanomaterials that can aid in smart biosensing. The review conducted by Chauhan et al., 2017 [14] concentrated on Graphene, its electrical activity, functionalization and other characteristics as well as its potential use in biosensing technology for detection of diseases. However, the review does not cover POC diagnosis, IoMT, and AI. Sadique et al. [15] provided a review on graphene based IoT integrated electrochemical biosensors for rapid diagnosis of SARS-CoV-2. Despite close similarities with the current review, the study does not cover broad topics and open research issue. The review provided by Soni et al. [16] is limited to the application of a carbon-based electrochemical biosensor for the detection of SARS-CoV-2. The review covers several aspects such as graphene, electrochemical biosensors and nanobiosensors, and AI-based detection. However, some of the dissimilarities with this review is the limitation on COVID-19, the absence of IoMT-based platforms, and open research issues.

The summary for the comparison with similar study is presented in Table 1.

### 1.2. Scope

This review paper aims to overview the potential of a nano-graphene-based electrochemical biosensor integrated with IoMT for point-of-care diagnosis of diseases. In order to hypernet between the research keywords, first we provide background knowledge on the definition of biosensors and its components, which include samples, bioreceptors, and transducers. Subsequently, we present the classification of biosensors in electrochemical, electrical, optical, piezoelectric, and thermometric properties and application of biosensors in clinical diagnosis, food control and monitoring, and environmental monitoring. Moreover, we provide an overview on graphene, its conductivity properties, and its preparation techniques. Nexus is the definition of nanotechnology, classification of nanoparticles with a sole focus on nanocomposite, their synthesis, application, and advantages. We also discuss the conjunction between electrochemical biosensors and nanomaterials supported by the literature and propose future nanographene-based electrochemical biosensors for smart points of care diagnosis.

The remaining part of this article is organized as follows. Section 2 overviews biosensors, components of biosensors, and classification and application of biosensors in clinical diagnosis, food control and monitoring, and environmental monitoring. In Section 3, we discuss graphene as a material in addition to its properties, application, and preparation. In Section 4, we link electrochemical biosensors with nanomaterials with support from the literature. In Section 5, we highlight several supporting research studies and propose a future biosensing system that incorporates graphene nanomaterials, AI, and IoMT for diagnosis of POC. In Section 6, we outline current open research issues regarding synthesis and fabrication of nanobiosensors, point-of-care diagnosis, and data privacy. Lastly, we provide concluding remarks in Section 7.

## 2. Biosensors

Biosensors are termed as analytical devices which are composed of biological compounds (e.g., microbes, enzymes, nucleic acid, whole cell, cell organelles, cell receptors, and antibodies), biologically derived and modified materials (e.g., recombinant antibodies, engineered protein, and aptamers), or biomimetics (e.g., ligands, synthetic catalyst, imprinted polymers, etc.) coupled in a physical transducer (e.g., optical, piezoelectric, magnetic, electrochemical, thermal, etc.). The first concept of functional biosensors is dated back to the study conducted by Clark in 1962 on detection of glucose using glucose oxidase (GOx) and oxygen electrodes as a transducer. The ideal characteristics of biosensors include high specificity, delicacy, versatility, shorter recuperation time, etc., [17,18,19,20]. Figure 1 shows the components of biosensors including analyte, transducers, and processors. Table 2 presents the advantages and disadvantages of different types of biosensors.

### 2.1. Components of a Biosensor

(A)Samples: Samples used in biosensors vary from human samples such as saliva, urine, blood, and other bodily fluid-to-cell cultures of bacteria and fungi and other microbes. Other types of samples include food samples and environmental samples such as water, soil, vegetation, etc.(B)Bioreceptors: Receptors used in biosensors differ from each other in terms of their mechanism, the most widely used bioreceptors include nucleic acid (NA) such as DNA and RNA, enzymes, antibodies, whole cells, cell organelles, immunosystems, tissues, hormones, lectin, etc.(C)Transducers: The function of a transducer is to transfer a signal from output domain detected from the chemical reaction between analyte and biological recognition element. Transducers are also called sensors, electrodes, or detectors. Transducers used in biosensors include thermal, optical, electrical, electrochemical, and piezoelectrical detection [19].(D)Electronics and Display: this component of a biosensor is responsible for processing and preparing the transduced signal for display. It is made up of a complicated electrical circuit that performs signal conditioning functions such as amplification and digital signal conversion. The display device of the biosensor then quantifies the processed signals [19].

In order to modify all types of biosensors, scientists incorporate NPs to enhanced desired properties such as sensitivity, flexibility, reliability, etc., [21]. Different types of NPs and NCs are utilized which include liposomes [22], gold NPs [23], dendrimers, CdSe/ZnS core shells for optical biosensors [24], and paramagnetics and supermagnetics for magnetic biosensors [25]. Gold NPs are also integrated on screen printed carbon electrodes [26], single walled carbon nanotubes [27], and graphite epoxy composites [28] for designing biosensors.

### 2.2. Classification of Biosensors

Biosensors can be categorized according to the mode of signal transduction (such as optical, thermal, electrochemical, magnetic, piezoelectrical, and electrical), biologically-specific countering mechanisms or number of usages as shown in Figure 2, for example, a single use (i.e., biosensor device that is disposable after a single measurement and thus unable to monitor or continue to analyze concentration after use) and a multi-use biosensor device, which is the one that can rapidly measure analyte continuously [29]. This article is only limited to electrochemical biosensors due to their wide applications in clinical diagnosis.

### 2.3. Electrochemical Biosensors

Definitions, nomenclatures, and classifications of electrochemical biosensors have been formulated by the International Union of Pure and Applied Chemistry (IUPAC) and its Analytical Chemistry and Physical Chemistry Divisions (i.e., Commission 1,7 and V.5 on Biophysical Chemistry, formerly the Steering Committee on Biophysical Chemistry and Analytical Chemistry). An electrochemical biosensor is characterized as a self-contained integrated tool in respect of these platforms that has the ability to provide specific quantitative or semi-quantitative analytical data or information through the use of biological recognition elements such as biochemical receptors that are designed and retained in a direct spatial contact with electrochemical transducer [29,30].

Understanding of the basic principle behind atoms, elements, compounds, and their reactions allow scientists to develop several applications ranging from diverse chemical products, medical procedures, fuels, batteries, etc. The field of theoretical chemistry continues to unravel mysteries surrounding chemical compounds, their constituents, properties, reactions, etc. Over the years, scientists have conducted several experiments supported by theoretical chemistry, which led to the development of several principles, products, and inventions. The field of electrochemistry was revolutionized by Faraday’s two laws which include: (1) the amount of a substance deposited on each electrode of an electrolytic cell is directly proportional to the quantity of electricity passed through the cell and (2) the quantities of different elements deposited by a given amount of electricity are in the ratio of their chemical equivalent weights [31,32,33].

The fundamental process of electrochemistry is the transfer of electrons between electrode surfaces and molecules in solutions. It is a powerful tool to prove reactions involving electron transfers, which relate the flow of electron to chemical changes. The fundamental mechanism behind electrochemical biosensors revolves around the chemical reaction between immobilized biomolecule and target analyte consuming or producing electrons or ions which in return affect measurable electrical properties of the solution (e.g., electric current and potential) [31]. Amperometric and potentiometric types are the major types of electrochemical biosensors.

(A)Amperometric Electrochemical Biosensors

Amperometric electrochemical biosensors are the most popular biosensors globally. One of the advantages of this class of electrochemical biosensors over potentiometric electrochemical biosensors is that they are suitable for mass production and are very sensitive. The mechanism behind this type of biosensor is based on amperometric quantification or detection of biochemical compounds via enzyme-catalyzed phosphorylation/hydrolysis or enzyme-catalyzed electroreduction or electrooxidation trailed by their usage in bioaffinity reaction, enabling electroreduction/electrooxidation [19,29,31].

Electrodes (such as a working electrode (WE), reference electrode (RE), or counter electrode) potentiostat and electrolyte are part of the electrochemical biosensor system as shown in Figure 3. WEs are micro-electrodes that carry out fascinating electrochemical events. A potentiostat is used to monitor WE’s applied potential as a function of RE potential, in the potential range of interest. Glassy carbon, Pt, Hg, and gold are some of the examples of WE. RE has a well-defined and stable equilibrium potential, using an electrochemical cell as a reference point to which the electrode’s potential can be measured. Normal Hydrogen Electrode (SHE), Calomel Electrode, and AgCl/Ag Electrode are examples of these electrodes. The circuit is completed by CE; it conducts electrons through the solution to WE from the signal source. Hg and Pt provide examples of these electrodes [32,34].

Application of a known potential and monitoring the current are the main objectives of the potentiostat. A simple potentiostat circuit with a three electrode and three operational amplifiers is most commonly used. Salt is the common supporting electrolyte, and it should be highly soluble in solvent [32,34]. Currents are generated as a result of applying potential, and the resulting alteration of electroactive species in the enzyme layer of the WE is measured [35,36].

(B)Potentiometric Electrochemical Biosensors

Currently, there are many potentiometric biosensors available commercially, such as metal oxide-based biosensors, glass electrodes, and ion-selective electrodes. The mechanism of a potentiometric electrochemical biosensor is the use of a transducer to determine that the distinction is plausible and which is obtained using diagonally ion-selective tissue placed in order to serve as a barrier between two solutions without current flow. This type of biosensor can be easily fabricated in large quantity in their miniature formats by employing advanced modern thick-film or silicon technological approach [19,37].

### 2.4. Application of Biosensors

Biosensors have numerous applications ranging from within the food industry to clinical diagnosis to environmental monitoring. The applications of biosensors in each sector are discussed below:

Clinical Diagnosis: The field of biosensors is rapidly growing as a result of technological advancement in electronics, as well as in nanotechnology and discoveries in biology and molecular medicine. However, its application in clinical diagnosis is hampered by many challenges, such as interference with undesired molecules during the diagnosis stage using human samples as well as low accuracy, selectivity, and specificity. Currently, 90% of manufactured biosensors related to healthcare are glucose biosensors. Other biosensors used in medical settings include detection of creatinine, cholesterol, triglycerides, and lactate [38,39,40].

Food Control and Monitoring: The application of biosensors in food sectors and biotechnology is not as common compared to medical or clinical diagnosis. In the medical care system, the major samples utilized for detection using biosensors include serum, blood, and urine, while there are no exact sample types employed in the food industry due to different constituents of biomaterials and biocomponents with variable compositions. Nowadays, there are amperometric biosensors based on oxidoreductase enzymes developed for detecting analytes such as fructose, glucose, glycerol, malic acid, acetic acid, and lactic acid in beverages and wines [31].

Environmental Monitoring: Due to current global warming and high emission of greenhouse gases from industries, monitoring of air, land, and water pollution is crucial. Treatment of water is among the techniques that undergoes modifications in the last century. However, the detection of toxic chemicals, waste water effluents, and heavy metals is critical for the prevention of several diseases. Currently, several biosensors and bioassays have been fabricated for evaluation of toxic compounds in water samples. Other biosensors developed include those capable of detecting pesticides and other toxic compounds [19]. The application of biosensors is shown in Figure 4.

### 2.5. Advantages of Electrochemical Biosensors

The application electrochemical biosensors continues to grow due to their prospects in clinical diagnosis, environmental monitoring, and food processing. The field of electrochemical-based biosensors is growing exponentially due to electrochemical-based biosensors’ advantages over other types of biosensors in addition to the integration of nanomaterials such as gold NPs, graphene and its derivatives, quantum dots, fullerenes, and other carbon allotropes. Some of the inherent advantages of electrochemical biosensors include their excellent detection limits, easy miniaturization and easy miniaturization (ease of fabrication), robustness, their ability to be utilized in biofluids with optically absorbing and fluorescing features, and their evaluation using conveniently drawn samples also known as liquid biopsies while additionally and simultaneously solving most of the limitations of contemporary approaches to accomplish a higher level of performance (such as a faster detection, sensing low amount, or concentration of target. As reported by Anik et al. [41], electrochemical biosensors offer high sensitivity, practicality, and fast responses. These attributes make them suitable as lab-on chips for point-of-care detection. Electrochemical biosensors are currently in use for detection of a wide array of biomolecules present in food samples and human body such as glucose, DNA, lactate, hemoglobin, alanine aminotransferase, uric acid, blood ketones, acetylcholine, cholesterol, cardiac troponin, etc.

## 3. Graphene

### 3.1. Carbon-Allotropes

In the last few years, we have witnessed the increased application of carbon-allotropes in biomedical sensing. Carbon-allotropes come in different varieties which include diamonds, fullerenes, lonsdaleite, graphite, different forms of graphene, nanotubes and nanohorns etc. Carbon-allotrope nanomaterials are currently used in several applications due to their unique properties. Some of these applications include biosensing, drug delivery, tissue engineering, bioimaging, medical diagnostic, and cancer therapy [42].

In the field of biosensing technology, carbon-allotropes are used extensively due to their inimitable properties including their electrical, optical, and mechanical properties as well as their flexibility, thermal stability, high electron mobilities, and strength-to-weight ratio. These properties make them suitable for miniaturizing sensors with low power drainage and high performance. Moreover, apart from their high sensitivity, carbon-allotrope-based biosensors offer higher special resolution in regard to localized detection along with POC and non-destructive and label free sensing. Some of the recent carbon allotropes reported in the field of biosensors include carbon nanotubes, fullerene, quantum dots, graphene, and its derivatives (such as reduced graphene and GO), nanodiamonds, etc. [42,43].

### 3.2. 2D Graphene

Graphene is a 2-dimensional form of crystalline carbon which exists as either a single layer of carbon atoms known as hexagonal lattice (i.e., which formed a honeycomb structure) or many coupled layers. The word graphene, when mentioned without indicating the form (i.e., multilayer or bilayer), mostly refers to a single layer graphene. Graphite, in contrast, is a three-dimensional form which is composed of moderately graphene layers. All graphitic structures are derived from graphene, which is known as the parent form. The forms of graphene exist as scrolls known as nanotubes, whereas spherical forms are known as buckyballs, which are the compounds of hexagonal and pentagonal rings of graphite [44,45].

The first studies conducted for understanding the electronic structure of graphite date back to 1947. The term graphene was first used in 1986 by a group of scientists known as Ralph Setton, Hanns-peter Bochm, and Eberhard Stummp. The name came as a result of combination of graphite depicting crystalline form of carbon while the suffix “ene” depicting polycyclic aromatic hydrocarbons (6-sided carbon atoms or hexagonal ring structure). However, it was not until 2004 that a simple technique by Konstantin Novoselov and colleagues known as exfoliation was used to isolate graphene from graphite. The method is also known as “the scotch tape” technique because it is based on the use of adhesive tape to withdraw the top layers of graphite and pass the removed layers to substrate material [46].

### 3.3. Properties and Applications of Graphene

The two-dimensional structure is of great interest to scientists in the field of chemistry, physics, biology, and sub-fields such as nanotechnology and electronics. A solid material which is composed of only a single layer of atoms arranged in an orderly manner is one of the examples of a 2-D crystalline form of graphene which exists as interfaces, surfaces, and membranes, offering a wide application in different fields of science. In computer engineering, chips are developed based on their integrated circuit due to their semiconducting properties. Graphene is considered to be among the valuable compounds with a potential application in electronics such as in transistors and other electric devices. The diverse properties of graphene include flexibility, strength, conductivity (electrically), and transparency, and it can be employed as a compound for fabrication of the torch screen. Another vital property of graphene is thermal conductivity which can be harnessed to remove heat coming from an electronic circuit [45,47]. In biology, the strength properties of graphene can be of advantage for use as a scaffold for understanding biological materials and molecules [48,49,50].

### 3.4. Conductivity of Graphene

The accomplishment of the exfoliation method has led scientists to understand different characteristics of graphene and its physical properties. The research also shows that electrons present in graphene possess an active mobility which opens up several uses of graphene in the field of electronics.

The fundamental electronic structure of graphene and its electric properties are very unique compared to other compounds. The use of chemical doping to absorb atoms and molecules using gate voltage can lead to the formation of a hole or electron. The formation of a hole is attributed to the missing electron in a specific region which acts as a positive electric charge. The conductivity of graphene is associated with conductivity formed in semiconductors as the compound does not have an insulator state. Therefore, conductivity remains finite even at zero doping [14].

### 3.5. Preparation/Synthesis of Graphene

Over the years, scientists have developed several approaches for the production of graphene. These methods range from the exfoliation method, which is expensive, to less expensive approaches such as the production of graphene from food, insects, and waste as well as the synthesis of graphene from natural and industrial carbonaceous wastes. The production of graphene proposed by Geim and Novoselov is based on covering graphene with a thin transparent layer of silicon dioxide (SiO_2_). The single layer of graphene formed based on this technique is visible under a standard optical microscope. The optical contrast created as a result of silicon dioxide increases the strength of the compound. The visibility of graphene under the microscope has two effects. Firstly, the optical contrast which is strongly enhanced by the interference phenomena in the layer covered using SiO_2_ causes the formation of rainbow colors in films of oil on water or soap film. Secondly, the strong interaction of graphene electrons and photons of visible light frequencies lead to absorption of approximately 2.3% of light intensity per atomic layer [51].

The high cost of this technique limits its usage. Thus, progress in the development of new techniques for industrial scale is highly required. However, since the study conducted by Geim and Novoselov in 2004, several methods have been proposed such as the production of graphene layers through the burning of silicon carbide abbreviated as SiC (a very hard compound produced synthetically from crystalline silicon and carbon). Another method is through chemical vapor deposition of carbon on the surface of some metals (i.e., nickel and copper) [51].

The study proposed by Stankovich et al. [52] revolves around the treatment of GO with Organic Isocyanates which can be exfoliated into functionalized graphene oxide nanoplatelets. Akhavan [53] proposed the production of graphene thin films with very low concentration of oxygen-containing functional groups. These graphene thin films are produced via the reduction of GO nanosheets (i.e., produced via chemical exfoliation) in a reducing medium as well as one and two-step heat treatment methods. Cornor- Márquez et al. [54] proposed the production of graphene flakes, multi-layered flat graphitic structures (nanoribbons), and partially opened multi-walled carbon nanotubes via the intercalation of lithium and ammonia followed by exfoliation of multi-walled carbon nanotubes. While the study proposed by Akhavan [55] for the first time demonstrated the synthesis of remote controllable working graphite nanostructured swimmer based on graphite het nanomotor.

## 4. Nanotechnology

### 4.1. Definitions and Properties

Nanotechnology is one of the most promising fields which has evolved due to transformation of science and technology [56]. Nanoparticles (NPs) are defined as materials with dimensions in nanometer (nm) range (1 nm is equivalent to 10^−9^ m). NPs can be found in nature or can be synthesized in the laboratory. Due to their smaller or infinite size, they possess unique properties that contribute to their immense application in the fields of engineering, medicine, and the environment. According to the International Organization for Standardization (IOS), NPs are defined as nano-objects in which all the 3 cartesian dimensions are below 100 nm. Nanoplates and nanodevices are two-dimensional while nanotubes and nanofibers are one-dimensional. The definition is further modified and expanded by the Commission of European Union (CEU) in 2011 as a nano-object with one or more of its dimensions ranging between 1 to 100 nm regardless of other dimensions being larger than 100 nm. The properties of NPs include physical, chemical, magnetic, optical, electrical properties, etc. The physical properties of NPs include high surface area and mobility (i.e., they are highly mobile), and they all exhibit quantum defects [57].

### 4.2. Classification of NPs

There is no exact classification of NPs. However, scientists categorized them based on material properties, size, chemical constituents, and shapes. In terms of chemical nature, NPS are subdivided into inorganic (such as gold, fullerenes, and quantum dots) and organic (such as liposomes, dendrimers, polymeric NPs, etc.). Other classifications of NPs are based on conductivity (conductors and semiconductors), carbon-based material composition (such as polymeric, metallic, ceramic, etc.), states such as soft NPs (nanodroplets, liposomes, vesicles), hard NPs (fullerenes, silica or silica dioxide, titania or titanium dioxide), and applications such as in medicine (therapy and diagnosis), electronics (molecular electronics), imaging (quantum dots), catalysis, magnetism, etc. [58]. The classifications of NPs are presented in Table 3.

### 4.3. Nanocomposites

Nanocomposites are described as hybrid or heterogenous materials formed as a result of the combination of polymers with inorganic solids within the nanometric scale (Sen, 2020). A composite is described as a solid material which occurs naturally or is synthesized as a result of a mixture of two or more materials with distinct composition and significant characteristics (chemical and physical) to form a new material with enhanced or desirable properties in comparison with each material separately [59,60].

### 4.4. Synthesis of Nanocomposites

Nanocomposites are produced via process of in situ growth as well as polymerization technique based on combination of inorganic matrix and biopolymer. Nanocomposites can also be synthesized by blending or combining metals, organometallic compounds, organic polymers, inorganic nanoclusters, clays, enzymes, fullerenes, biological compounds, etc. The wide application of nanocomposite revolves around the utilization of building blocks based on nanometer range dimensions to synthesize new materials with unprecedented enhancement or improvement in both chemical and physical properties [61].

### 4.5. Classification of Nanocomposites

The general classification of nanocomposites is according to the absence and presence of polymeric material as one of the constituents.

Polymer-based nanocomposites are copolymers or organic polymers which contain either nanofillers or NPs dispersed in their matrix which are often called poly nanocomposites. The criteria for regarding a compound as polymer-based nanocomposite is one of the dimensions must be within 1–50 nm. These nanocomposites come in different shape such as fibers, platelets, and spheroids [62]. In research development and application, polymer-based nanomaterials are utilized more compared to other classes of nanocomposites. These types of improved or enhanced material have shown characteristics such as dimensional variability, activated functionalities, and film forming ability among others [63].Non-polymer-based nanocomposites are nanocomposites in which one of the constituents does not comprise of any organic polymer or polymer derived materials. This type of nanocomposite is also called an inorganic nanocomposite. Their subclassifications include nanocomposites based on ceramics, and metallic nanocomposites [64].

### 4.6. Advantage of Nanocomposite

The formation of nanocomposites as a result of blending 2 or more separate building materials or constituents provides desirable characteristics because of their large surface area and the interfacial relationship between the phases and small scale. In terms of application, many catalysts, drugs, biomaterials, etc., are commonly used to enhance the biological ability of nanocomposites. The science behind the development of composites is to provide a wide range of features and characteristics, such as resistance to corrosion, resistance against fatigue, strength and stiffness, low coefficient of expansion, and simple repair of damaged structure [65].

## 5. Hyphenation

### 5.1. Electrochemical Biosensors and Nanomaterials

Nanomedicine is defined as the integration of nanotechnology in medical applications such as diagnosis, therapy, and treatment. According to European Science foundation (ESF), nanomedicine is defined as the application of nanometer sized tools for different biomedical applications such as diagnosis, prevention and treatment of diseases, and understanding the complex underlying pathophysiology for the purpose of improving the quality of patient’s life [66].

The application of nanotechnology in diagnosis of disease revolves around their extraordinary chemical and physical properties. Nanomaterials exist in different forms such as organic, metallic, or semi-conducting materials enhanced with chemical creativity and high surface volume-ratio. Nanoscale structures can be engineered in different shapes, sizes, and chemical compositions, which makes them suitable for clinical diagnostics [67,68]. The blending of biosensors with nanotechnology can improve real time analysis, low limit detection, high-throughput screening, less sample to volume requirements, and label-free detection [69,70].

The hyphenation between nanotechnology and clinical diagnosis is attributed to the pathology of many diseases which occur as a result of biological alterations at the nanoscale level (such as bacterial or viral infections, low abundance of protein, and mutated genes). Nanostructures could be engineered as carriers or agents to help in the transportation across biological barriers, mediate in molecular interactions, or gain access to molecules [71]. Moreover, nanomaterials exhibited unique electron with desired optical, electrical, and magnetic properties which aid in detection of analyte at molecular level. Additionally, small dimensions of these nanostructures can be exploited for device miniaturization and integration. The application of nanotechnology in diagnostics revolves around the development of biosensors and analytical devices for in vitro and in vivo modalities [72].

Chowdhury et al. [73] proposed a pulse-triggered ultrasensitive electrochemical biosensor contrived by utilizing gold-embedded polyaniline nanowires and quantum dots prepared using interferential polymerization and self-assembly method. The sensor was designed to detect the hepatitis E virus from fecal samples obtained from an infected monkey cell culture. In terms of detection with the Real-Time Quantitative Reverse Transcription-Polymerization Chain Reaction (RT-qPCR) process, validation of the biosensor has shown similar sensitivity.

By using Nano-Porous Pseudo Carbon Paste Electrode (Nano-PPCPE) as WE coupled with other components such as enzymes (catalase (Cat) and glutamate oxidase (L-GLOD)), bovine serum albumin (BSA) and phosphate buffer (PB) with pH = 7.4 and subsequently cross-linked with glutaraldehyde, Deng et al. [74] prepared a novel L-glutamate electrochemical biosensor. The sensor’s assessment output reached a detection limit of 2.5 × 10^−7^ M with a linear range from 5 × 10^−7^ M to 1 × 10^−5^. Nano-PPCPE has demonstrated higher selectivity and sensitivity compared to modified carbon paste electrode (CPE).

Tian et al. [75] designed a simple label-free electrochemical biosensor for detection of microRNA-21. The sensor was designed using Gold (Au) NPs superlattice as a supporting material and Toluidine Blue (TB) as a redox indicator. The combination of these two compounds have shown to induce remarkable effects on sensor’s signal amplifications. The gold NPs were coated using polypyrole and conductive polymer which was later self-assembled to produce superlattice capable of producing stream. Discrepancy Pulse Voltammetry (DPV) and Cyclic Voltammetry (CV) are utilized to confirmed the hybridization between Single Stranded RNA (SSRNA) probe immobilized on the electrode and the target microRNA sequence as well as determination of oxidation peak current of TB under optimal condition of pH, working potential and concentration. The system was able to detect microRNA between the range of 100 aM to 1 nm and 78 aM low detection limit.

Liu et al. [76] developed an aptamer-dependent electrochemical biosensor updated with aptamer, gold NPs, and chitosan for the detection of cadmium ions (Cd2^+^) based on the use of Glass Carbon Electrode (GCE). The primary mechanism of the sensor is based on the use of poly (diallyl dimethyl ammonium chloride) also known as PDDA applied to neutralize cd-aptamer as a result of the presence of Cd2^+^, causing the aptamer complex to absorb higher electrochemical signal indicator than PDDA due to electrostatic interaction triggering cd-aptamer conformation shift. The sensor performance was evaluated using DPV resulting in 0.001 nm to 100 nm linear ranges and low detection limit of 0.04995 pM.

An enzyme-free electrochemical biosensor was developed by Zhang et al. [77] in order to accurately detect the species Pseudomonas aeruginosa (i.e., most interactable multidrug resistant bacteria). The biosensor was manufactured using the zirconium-based metal-organic system (ZrMOF) synthesized zirconium series, which has desirable characteristics such as excellent absorption and large surface area. The compound was linked to Cu2^+^ to form Cu-ZrMOF that exhibits elevated catalytic activity. The primary mechanism of the electrochemical biosensor revolved around the utilization of ZrMOF-aptamer-DNA nanocomposite as signal probe to catalyze H_2_O_2_ decomposition. The detection ability of the sensor is within linearity range of 10 to 106 CFU ML^−1^ with a low detection limit of 2CFU ML^−1^. Furthermore, the developed sensor has shown remarkable quantification of Pseudomonas aeruginosa in samples of spiked urine.

Cinti et al. [78] proposed a blood glucose electrochemical biosensor based on Prussian Blue NPs. The “paper blue” is a functional sensor developed via integration of PBNPs on filter paper using its precursors (in µL concentration) in the absence of external inputs (e.g., reducing agents, voltage, and pH). To generate a reagentless electrochemical biosensor that can be used as a point of treatment and self-monitoring for diabetes, the paper blue is combined with screen-printing and wax. The working mechanism of the electrochemical biosensor rely on the sensing ability of the paper blue, exploiting the reduction of H_2_O_2_ at applied potential using GOx as the biological recognition element. The linearity detection of the sensor ranges up to 450 mg/dL (25 mM).

Zhao et al. [79] developed a high-performance fiber-shaped wearable sensor for electrochemical detection of glucose in sweat samples. The biosensor was fabricated using 3 electrodes: (1) working electrodes based on functionalization of gold fiber with GOx and Prussian blue, (2) reference electrode based on modified Ag/AgCl, and (3) counter electrode based on nanomodified gold fiber. The strain-insensitive and highly stretchable fiber-based wearable electrochemical biosensor achieved 0–500 µM linear range and 11.7 µA mM^−1^ cm^−2^ sensitivity.

Hajian et al. [80] developed a CRISPR-Chip biosensor using graphene-based field effect transistor (gFET) which utilized the recent and buzzing genetic engineering system known CRISPR for digital detection of target sequence from HEK293T cells of patients suffering from Duchene Muscular Dystrophy (DMD). The working mechanism of the CRISPR-Chip biosensor is based on DNA-target ability of dCas9 (deactivated form of CRISPR associated system 9). The sensor was able to achieved 1.7 fM sensitivity within 15 min without the need target sequence amplification.

Varmira et al. [81] considered an ultrasensitive enzymatic biosensor for the detection of tyrosine in food samples (yogurt, cheese, egg, etc.). The biosensor was fabricated based on immobilization of tyrosine hydroxylase on electrode composed of GCE, Palladium-Platinum bimetallic alloy, graphene multi-walled carbon nanotubes, chitosan, etc. The immobilization process was carried out by cross-linking chitosan and tyrosine hydroxylase via the addition of glutaraldehyde on the surface of the biosensor. The sensor’s working theory is based on the selectivity and affinity of catalytically active proteins on target molecules where tyrosine hydroxylase directly catalyzed the conversion process from tyrosine to levodopa, which oxidized at a lower potential compared to tyrosine (which oxidized at a higher potential). Biosensor characterization has been carried out. However, under optimal conditions of pH, concentration, and working potential, the biosensor detected tyrosine within 8.0 × 10^−9^ to 160.0 × 10^−9^ mol L^−1^ and 0.01 × 10^−9^ to 8.0 × 10^−9^ mol L^−1^ concentration ranges and a 0.009 × 10^−9^ mol L^−1^ limit of detection.

### 5.2. Graphene-Based Biosensors

The emergence of graphene as a 2D nanomaterial has transformed the field of electronics and sensing technology due to their wide properties such as stable under ambient circumstances, ultra-high charge mobility, semi-metallic characteristics, and unique electronic band structure that contributes to their outstanding electronic properties. The use of graphene, subtypes, and functionalize types such as quantum dots graphene, GO, and RGO are widely used in biomedical applications such as biosensing and bioimaging. Their application in medicine is attributed to their large surface area, ability to interact with wide array of biomolecules, and other desirable properties which include biocompatibility, solubility, and functionalization [82,83,84].

Among the existing nanomaterials, graphene has of the largest surface areas which contribute to their applications in biosensing as a result of interaction with biomolecules. Their applications in biosensing can be categorized into 2: (1) immobilization approach based on chemical functionalization of the surface of the graphene nanomaterial (such as GO, rGO, and graphene-quantum dots) with molecular receptors, and (2) the second approach involves charge biomolecule interactions at π-π domain, electrostatic forces, and change exchange resulting in electrical variations in the pristine graphene [85].

The existing literature has overviewed the structure, the extraordinary properties and preparation of graphene-based materials for applications in biosensing technology. The pristine graphene has shown to offer infinite surface at molecular level, high electrostatic force, noncovalent interactions, and π–π stacking. Thus, graphene as a material with high surface area and active sites provides high prospects for charge molecular interactions which contribute to their applications in biosensing technology and improvement of desired characteristics such as selectivity and enhanced sensitivity [86]. The improvement of performance, LOD, and sensitivity of graphene in biosensing applications is attributed to the improvement of charge electron transfer between biomolecules and graphene. Moreover, the chemical functionalization of graphene allows interactions with several biomolecules and compounds such as nucleic acid, antibodies, antigens, enzymes, quantum dots, nanoparticles, heteroatoms, etc., [87,88].

In order to detect lead in blood samples of children, Wang et al. [89] developed a portable free-label aptamer biosensor using graphene field effect transistor gFET. The aptasensor was designed based on the interaction between 8–17 DNAzyme and the Thrombin binding aptamer, and achieved a detection limit below 37.5 ng/L using a standard solution of several lead concentrations. The device was also tested on a real blood sample and has shown high sensitivity. In terms of selectivity, the device has shown good selectivity over other metals such as calcium, potassium, sodium, and magnesium. The use of label-free graphene-enhanced biosensors for the detection of cancer is proposed by Zhu et al. [90] The immunosensor is designed using gFET which is immobilized with antibodies based on non-covalent pi stacking. The device takes advantage of the interaction between immobilized antibody targeting carcinoembryonic antigen (CEA). The real time detection of CEA protein using the anti-CEA modified gFET achieved high selectivity and sensitivity, leading to LOD of less than 100 pg/ML.

Teengam et al. [91] developed an inexpensive and disposable electrochemical paper-based peptic nucleic acid biosensor for the detection of human papillomavirus. The biosensing mechanism revolves around surface modification of electrodes using graphene-polyaniline and immobilization of anthraquinone-label pyrrolidinyl peptic nucleic acid through electrostatic attraction. The biosensor was used to detect synthetic 14-base oligonucleotide target sequences which correspond to real human papillomavirus type 16 DNA. The resulting interactions were measured using square wave voltammetry. The nucleic acid-based graphene modified biosensor was able to achieved 2.3 nM limit of detection and 10–200 nM linear range. The developed sensor was also used to detect an amplified DNA target from a real sample.

Li et al. [92] developed a novel electrochemical aptasensor for the detection of cardiac troponin. The biosensor was designed based on the synergistic effect of three materials which include rGO, silver nanoparticles, and Molybdenum sulfide (MOS) for the stable immobilization of the aptamer. In order to analyze the developed biosensor, several electrochemical tests were conducted which include CV, DPV, and electrochemical impedance spectroscopy. The proposed electrochemical aptasensor has shown to detect samples within the linear range of 0.3 pg/mL to 0.2 ng/mL. Saeed et al. [93] developed an electrochemical DNA biosensor for the detection of breast cancer biomarkers. The biosensor was designed based on the modification of glassy carbon electrons with GO and gold nanoparticles and subsequent immobilization of 2 different DNA which include CD24c and ERBB2c. In order to measure the response of the amperometric detection of the biosensor, a sandwich-type detection strategy was employed. The biosensor achieved 0.16 nM and 0.23 nM detection limits and 378 nA/nM and 219 nA/nM for ERBB2 and CD24, respectively.

The construction of the biosensor based on the integration of platinum and graphene to form RGO-Pt nanocomposite for the detection of living cell peroxide was proposed by Zhang et al. [94]. The nanocomposite structure was modified on the surface of glassy carbon electrode using both adsorption and electrodeposition approach. The developed electrochemical biosensor achieved a 0.2 µM detection limit and a 0.5 µM to 3.475 mM wide linear range. Liu et al. [95] developed a highly specific and stable DNA-based biosensor for the detection of mycobacterium tuberculosis specific DNA sequence. The biosensor was developed using RGO- gold NPs as a sensing platform and gold NPs-polyaniline as a tracer label for amplification. The mechanism behind the detection approach of the developed biosensor revolves around the electrodeposition of gold NPs on the surface of the rGO modified electrode and subsequent immobilization of DNA probe. The DNA-based biosensor exhibited high sensitivity with a linear range of 1.0 × 10^−15^ to 1.0 × 10^−9^ M.

The study conducted by Akhavan et al. [96] applied reduced graphene nanowalls for the development of an ultra-high-resolution electrochemical biosensor for detection of the 4 DNA bases as well as monitoring the oxidation signals of the individual nucleotide bases (A, G, C, and T). Moreover, the study also compared the electrochemical performances of the modified electrochemical reactivity of the 4 DNA bases, single-stranded DNA, and double-stranded DNA (dsDNA) at the surface of the reduced graphene nanowalls electrode with glassy carbon, graphite, and reduced graphene nanosheets electrodes. The electrochemical performances were evaluated on the basis of DPV scans which led to the linear dynamic detection range of 0.1 fM to 10 mM using reduced graphene nanowalls for dsDNA and 2.0 pM to <10 mM using reduced graphene nanosheets, respectively. Moreover, the LOD of DsDNA for both reduced graphene nanowalls, and reduced graphene nanosheets were estimated as 9.4 zM (∼5 dsDNA/mL) and 5.4 fM, respectively. Comparison between the 2 electrodes in terms of SNP shows that reduced graphene nanowalls were more efficient with 20 zM oligonucleotides (∼10 DNA/mL).

Akhavan et al. [97] developed a fast ultrasensitive electrochemical biosensor for the detection of leukemia cells. The study employed reduced graphene nanowalls electrodes for the detection of leukemia cells at leukemia fractions (LFs) of ∼10^−11^ (as the lower detection limit) in a blood serum containing the extracted guanine. The electrochemical evaluation of the system was conducted using DPV and the signals acquired by oxidation of the extracted guanine on the reduced graphene oxide nanowall electrodes were evaluated after 20 cycles. The result shows that the DPV peaks relating to guanine oxidation (i.e., without guanine extraction) of normal and abnormal cells overlapped at LFs < 10^−9^. However, the DPV result using glassy carbon electrodes was able to detect only LFs ∼ 10^−2^. The result of this study has claimed that the ultra-sensitivity obtained as a result of the combination of both methods (i.e., guanine extraction by graphene oxide nanoplates and then guanine oxidation by reduced graphene oxide nanowalls) produced higher sensitivity than existing technologies.

The electrochemical detection of leukemia and normal blood cell using spongy graphene electrode was proposed by Akhavan et al. [98]. The biosensor was designed using Mg2+-charged spongy graphene electrodes (SGEs) which was fabricated using electrophoretic deposition of chemically exfoliated graphene oxide sheets on graphite rods. The electrochemical evaluation of the proposed biosensor using DPV resulted in linear dynamic detection behavior with wide concentration range of 1.0 × 105–0.1 cell/mL and LOD of ∼0.02 cell/mL. Other graphene-based electrochemical biosensors include identification of SARS-CoV-2 using dCas9 [99], determination of testosterone [100], determination of ascorbic acid, dopamine and uric acid [101], and detection of SARS-CoV-2 specific antigen using disposable electrochemical biosensor containing laser-induced graphene [102] and buffer-based zinc oxide/reduced graphene oxide nanosurface for the detection of SARS-CoV-2 antigens [103].

The summary of electrochemical nanomodified graphene-based biosensors is shown in Table 4.

### 5.3. Blending IoT and AI with Medical Devices

Technological advancement in computer science (internet, software, hardware) and medical device engineering are the major forces driving innovation in healthcare systems. Integration of IoT systems in healthcare is transforming the sector at an exponential rate. Coupling IoT and AI in medical devices has led to the development of smart system that can generate, collect, analyze, store, and transmit data. The merging of these technologies has the potential to help healthcare providers achieve better healthcare outcomes, improve efficiency (through increasing the reliability, accuracy, and productivity of medical devices), reduce healthcare error and cost, and provide new approaches of engaging and empowering patients [104].

The integration of IoT in biosensors have the potential to improve diagnosis and treatment of disease through early diagnosis and timely monitoring of diseases. IoMT also known as IoT-Healthcare is a branch of IoT that is concerned with the connection of medical devices and medical systems with healthcare professionals using the internet (online computer networks such as the Wi-Fi or Bluetooth or storing them in the cloud, biomedical data, or hospital records) [6,9].

The recent advancement in IoMT has led to several applications such as remote monitoring and tracking patient’s medications and tracking of patient within the hospital. Another application that improves healthcare delivery is the development of IoMT-wearable devices. These types of connected devices are capable of sending information from patients to healthcare professionals. Some of the IoMT-based devices developed include a wide range of wearable devices, hospital beds rigged with sensors that measure patient’s vital signs (such as heart rate and pressure), and infusion forms connected to analytical dashboards. In order to spare patients from regular visitations to the hospital and reduce cost, scientists developed “telemedicine” which utilize devices integrated with sensors that allow the monitoring of patients in the comfort of their own homes [6].

## 6. Proposed Future Graphene-Based Electrochemical Smart Biosensors

### 6.1. Challenges

One of the challenges facing diagnosis of diseases is the use of complex, time-consuming laboratory procedures. The current COVID-19 pandemic has increased the incidence of contagious diseases caused by different pathogens such as viruses and bacteria. To counter these challenges, there is high need for development of rapid, sensitive, specific, robust POCT which can ensure faster, on-site detection of diseases, which can be crucial for early and timely action by medical personnel [10].

### 6.2. Supporting Research

The integration of nanotechnology for the development of POCT has shown to improve signal detection and signal enhancement chemistries [105]. Several biosensing technologies have been developed for lab-on-chip devices which include piezoelectric, electrochemical, optical, and surfaced-enhanced Raman spectroscopy [106]. Despite the fact that POCT is an emerging technology, it has shown tremendous potential is improving detection of pathogenic diseases and management through timely monitoring and treatment [107,108].

Graphene has emerged as one of the ideal materials in biosensing technology due to their excellent dispersity [109], good electrical conductivity, ease of functionalization, biocompatibility, and optical and physiochemical properties [14]. In order to detect early Alzheimer diseases, Li et al. [110] developed a reusable biosensor using magnetic nitrogen-doped graphene (MNG) which is modified on gold (Au) electrode. The biosensor was able to achieve 5 pg mL^−1^ detection limit with linear range of 5 pg mL^−1^ to 800 pg mL^−1^.

### 6.3. Architecture and Framework of Proposed Biosensor

The most common biorecognition element used in biosensing include nucleic acid (DNA and RNA), enzymes, and antibodies. The majority of research in the literature that used graphene nanomaterials employed DNA as biorecognition element due to its specificity (hybridization between probe and target) as shown in the study conducted by Hajian et al. [80] who utilized CRISPR-Cas 9 (gene editing tool that cut target DNA) and Field Effect Transistor (FET). Yola et al. [111] utilized DNA as a biorecognition element in an electrical biosensor enhanced with gold-coated iron (Fe@Au) nanoparticles decorated with graphene oxide. Chen et al. [112] developed electrochemical DNA biosensor based on 3D nitrogen-doped graphene and Fe3O4 nanoparticles. Other studies that utilized DNA as a biorecognition element include Haung et al. [113], Chen et al. [114], Wang et al. [115], and Ye et al. [116].

Since the first development of the commercial glucose biosensor in 1975, scientists have developed several detection approaches that rely upon hybridization between target (NA) and probes labelled with fluorescent, chemiluminescent, and radioactive materials [117]. Apart from direct detection techniques, there are also indirect approaches that utilize enzymes to catalyze the generation of chemiluminescent, fluorescent, and colorimetric signal. Most DNA-based electrochemical or aptamer biosensors developed generated electrical signal due to redox reaction as a result of hybridization between target and probe. The electrical signal generated is measured or monitored using voltammetric techniques such as DPV, CV, Polarography, linear sweeps etc. [118].

In this section, we proposed an amperometric electrochemical biosensor that utilizes either nucleic acid, enzymes, or antibodies labelled with fluorescent, chemiluminescent or radioactive materials as a biorecognition element and the use of graphene nanocomposite immobilized on WE. The overall framework of the proposed biosensor is illustrated in Figure 5.

### 6.4. Sensing Technology and Signal Processing

The growing of developed biosensors enables detection of disease using different biorecognition element such as enzymes, DNA, antibodies, and whole cells. DNA-based electrochemical biosensors are generating interest due to their specificity (as a result of molecular hybridization between target DNA and probe). Integrating graphene nanocomposite into the electrode of biosensors increase conductivity. The proposed next-generation smart nanobiosensor will utilize nanographene composite integrated with biorecognition elements (DNA/RNA, enzymes, antibodies) labelled with fluorescent dye [119,120]. The molecular hybridization between either nucleic acid, catalytic reaction between enzymes and substrates, and binding interaction between antibodies and antigens will result in signal generation and color change. The electrical signal generated as a result of interaction between target and biorecognition elements will be translated by a signal processing unit into a digital readable output displayed on the biosensor screen.

### 6.5. Data Analysis and Transmission

The last two decades have seen the rise of application of AI-models and machine learning models for medical purposes. These models are used for data analysis of digital, graphic, and image inputs. For classification and prediction of digital inputs, several machine learning models such as clustering algorithms, logistic and linear regression, K-Nearest Neighbors (KNN), and Naïve Bayes are utilized [121]. The use of voltammetric electrochemical biosensors generate peak current and signal waves which can be stored in the form of images and classified using CNN or ANN models.

The result generated on the biosensor screen can be used as an input for AI-models for prediction or classification of severity of disease. The output of the resulting models can be transferred using wireless connection to healthcare professionals for easy decision-making or stored in the cloud system or hospital data storage system which can be accessible in the future or can be used by researchers for data analysis and other research purposes.

### 6.6. Challenges and Open Research Issue

A challenge faced by healthcare system is the emergence and reemergence of infectious diseases. On-site or point-of-care detection of these diseases is crucial for timely treatment and prevention. Despite the advancement in the field of biomedical engineering, nanotechnology, and electronics, the robust, rapid, sensitive, accurate, on-site or point-of-care diagnostics still remain a challenge. So far, progress has been made by scientists, most commonly towards the development of minimally invasive, contact (such as wearables) electrochemical biosensors for in vivo diagnosis [122,123].

A challenge related to enzymatic-based electrochemical biosensors is the issue of electron transfers which result from the biocatalytic reaction between immobilized enzymes and substrates [124]. Majority of the progress made towards enzymatic biosensors are related to glucose biosensors where ga lucose-oxidase (GOx) enzyme is immobilized on the electrode surface (third generation) unlike in the first generation and second generation glucose biosensors where GOx required cofactors such as flavin adenine dinucleotide (FAD) and mediators such as hydroquinone, ferricyanide, ferrocene, etc. Currently, scientists are exploring the use of nanoparticles to enhance the surface electrodes in order to improve conductivity and sensitivity [125].

Due to some of the constraints of enzymatic electrochemical biosensors, scientists explore the use of nucleic acid as a biorecognition element due to their specificity and sensitivity. Currently, DNA-based electrochemical biosensors are developed for rapid, sensitive, precise, and economical testing of infectious diseases using target nucleic and synthesized probe. Unlike enzymatic-based biosensors, nucleic acid-based biosensors can be readily synthesized and regenerated for multiple uses [126]. Another challenge facing electrochemical DNA biosensors is the need for amplification where target nucleic acid is amplified using PCR and other nucleic acid amplification (NAA) assays. However, the recent study conducted by Choi et al. [127], Vishnubhotla et al. [128] and Li et al. [129] have shown that detection can be achieved without the need for amplification (amplification free).

Most of the advancement made towards point-of-care DNA-based biosensors are based on optical transducers (such as plasmonic surface enhanced Raman spectroscopy). Integration of nanomaterials such as gold particles (AuNPs) and gold nanorods [130], nanofibers (Manganese (II), oxide [131], graphene and its composite (such as nanocomposite reduced graphene) also occurs [132]. Other nanomaterial use in biosensing technology includes dendrimers, nanowires, polymetric nanoparticles, quantum dots. Fullerene etc., [133] has shown to improve the functioning of DNA-biosensors.

One of the major concerns for healthcare providers regarding IoMT-assisted systems or devices is data privacy (security of patient data). The growing number of medical connected devices makes them prone to cyberattacks. In order to safeguard or secure patient data from malicious attacks as a result of machine– human and machine– machine interactions, data generated from biosensors must be equipped with strong privacy and security settings such as seamless authentication and authorization services (e.g., cryptographic technique used to encrypt and decrypt data) [134,135]. To avoid dubious attacks, future biosensors must be integrated with a smart security system that can protect data such as logs, raw data, metadate, etc.

## 7. Concluding Remark

Early detection of diseases is crucial for effective diagnoses and prevention of incidences of contagious disease caused by pathogens such as influenza virus, dengue virus, zika virus, human papilloma virus, SARS-CoV-1, SARS-CoV-2, MERS-CoV, HIV virus, malaria, etc. Despite the advancement made in the last few decades, conventional diagnoses are still hindered by many challenges which include low sensitivity, false positive results, time consumption, lack of point-of-care diagnostic devices, complex laboratory protocols, the need for expertise, etc. In order to solve the issue of low sensitivity and the need to increase conductivity detection chemistry and signal detection, scientists incorporate nanomaterials such as liposomes, gold NPs, and Cdse/ZnS dendrimer core shells for optical biosensors, magnetic biosensors (paramagnetic and super magnetic), screen-printed carbon electrodes integrated with gold NPs, single walled carbon nanotubes, non-porous film coated platinum NP, and electrical biosensor graphite epoxy composite Au NPs.

Development of POC diagnostic devices is highly required for on-site or real-time testing. This type of device is essential for timely treatment and prevention of emerging and reemerging infectious diseases. In order to explore these challenges, scientists need to incorporate IoMT which offer wireless-based connectivity of POC nanobiosensor with healthcare personnel and medical centers. Thus, in this review, we proposed a point-of-c are graphene-nanobiosensor integrated with IoMT (POCGNB-IoMT).

## Figures and Tables

**Figure 1 sensors-23-02240-f001:**
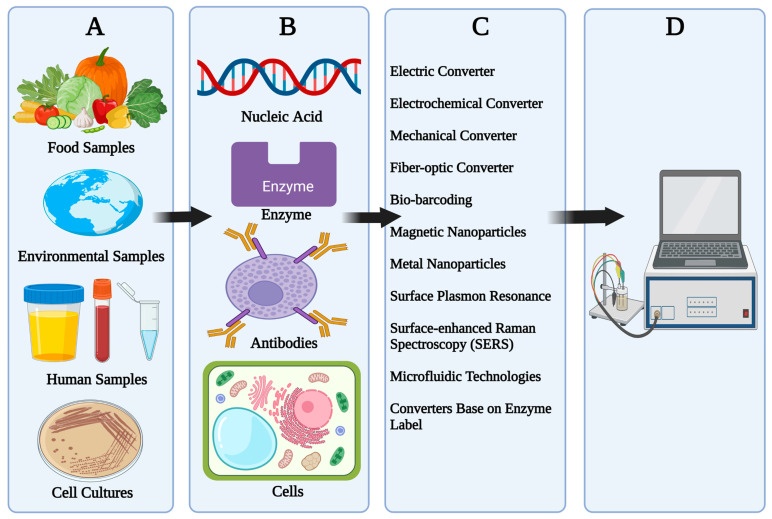
Components of a Biosensor. (**A**) Sample, (**B**) Biorecognition elements, (**C**) Transducer, (**D**) Display.

**Figure 2 sensors-23-02240-f002:**
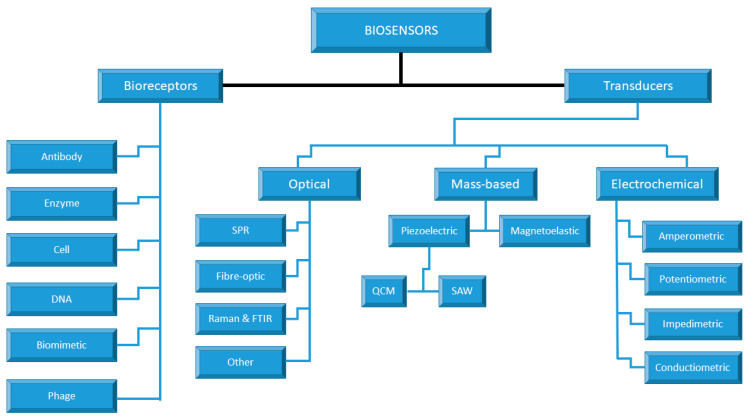
Classification of Biosensors.

**Figure 3 sensors-23-02240-f003:**
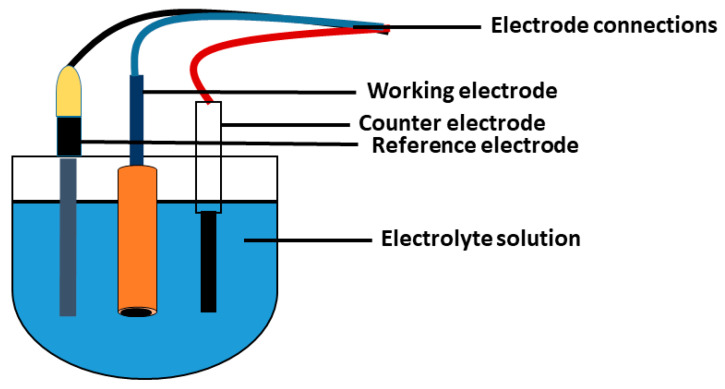
Electrochemical cell.

**Figure 4 sensors-23-02240-f004:**
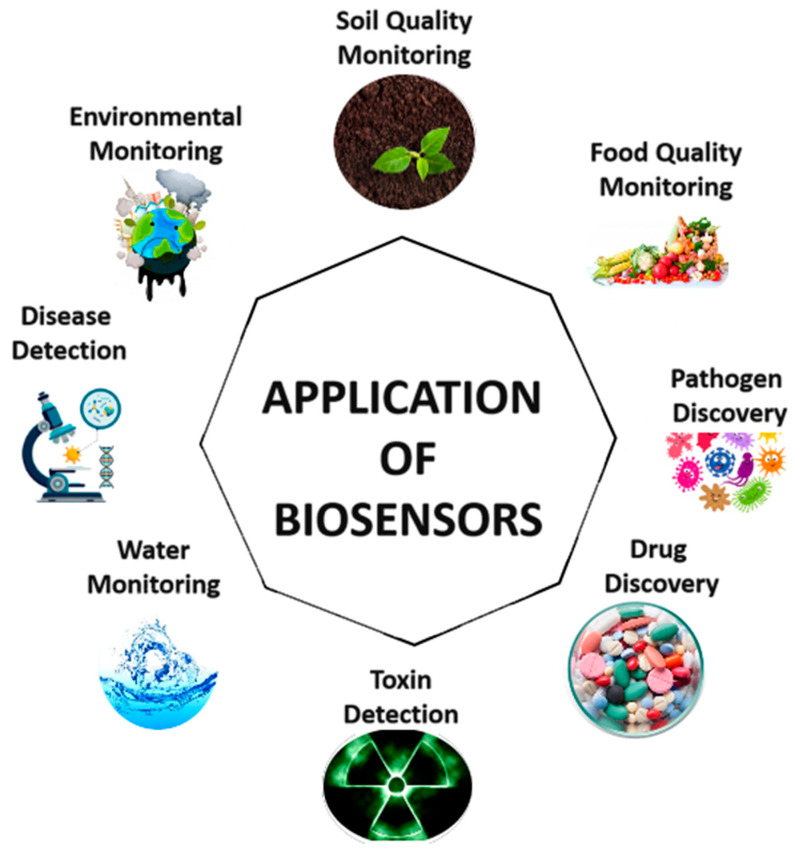
Application of Biosensors.

**Figure 5 sensors-23-02240-f005:**
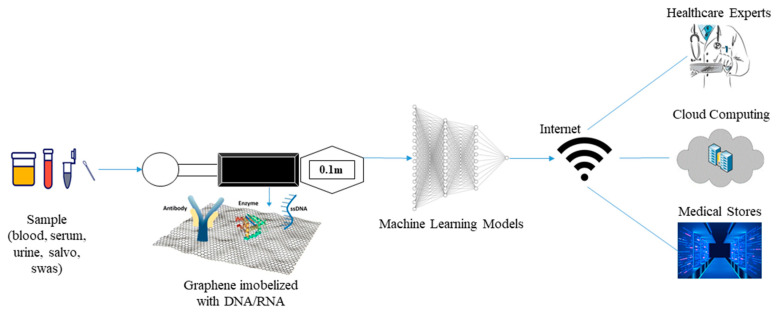
Proposed Biosensor.

**Table 1 sensors-23-02240-t001:** Comparison with similar studies.

Reference	Biosensors	Graphene Nanocomposites	Clinical Diagnostics	Medical Data, AI and IoMT	Open Research Issue
[11]	✓	-	✓	-	✓
[12]	✓	✓	✓	-	-
[13]	✓	✓	✓	-	✓
[10]	✓	-	✓	✓	✓
[14]	✓	✓	✓	-	-
[15]	✓	✓	-	✓	-
[16]	✓	✓	-	✓	-

**Table 2 sensors-23-02240-t002:** Advantage and Disadvantage of Different Types of Biosensors.

Biosensor	Types	Advantages	Disadvantages
Electrochemical	Amperometric, potentiometric, conductimetric, impedimetric and voltametric	excellent detection limits, easy miniaturization easy miniaturization (ease of fabrication), robustness, faster detection, linear output, good resolution, etc.	Unstable current and voltage, narrow or limited temperature range, cross sensitivity with other gases, short or limited shelf life
Optical	surface plasmon resonance (SPR), fluorescence, bio/chemiluminescence, refractive index, Raman scattering, absorbance	high sensitivity, selectivity, cost-effectiveness, small size	Susceptible to physical change and interference from environmental effects
Thermal/calorimetric	Thermistors or thermopiles	Scalability, Ease of use and ease of fabrication	Lack of specificity inn temperature measurements, long experimental procedures
Mass sensitive or gravimetric	Wave biosensor, surface acoustic, cantilever	Low cost and simplicity	Low sensitivity
Piezoelectric	Surface acoustic devices, Piezoelectric crystal	Fast detection, good frequency response, small size and high sensitivity	High temperature sensitivity, not suitable for static conditions, some crystal can dissolve in high humid environment and are water soluble

**Table 3 sensors-23-02240-t003:** Classification of Nanoparticles.

Classification	Example
Chemical Nature	Inorganic and organic
Conductivity	Conductors and semiconductors
Material composition	Polymeric, ceramic, metallic, etc.
NP state	Soft and hard
Application	Medicine, imaging (optics), electronics, etc.

**Table 4 sensors-23-02240-t004:** Electrochemical nanomodified graphene-based biosensors.

Reference	Sample/Analyte/Target	Type of Graphene	LOD/LR
[89]	Lead in blood sample	gFET	LOD below 37.5 ng/L
[90]	Cancer	gFET	LOD of less than 100 pg/ML
[91]	Human papillomavirus	graphene-polyaniline	2.3 nM LOD and 10–200 nM LR
[92]	Cardiac troponin	rGO, silver nanoparticles and Molybdenum sulfide (MOS)	0.3 pg/mL to 0.2 ng/mL LR
[93]	Breast cancer biomarkers	GO and gold nanoparticles	0.16 nM and 0.23 nM LOD and 378 nA/nM and 219 nA/nM for ERBB2 and CD24 respectively
[94]	Living cell peroxide	RGO-Pt	0.2 Um LOD and 0.5 µM to3.475 mM wide LR
[95]	Mycobacterium tuberculosis	RGO- gold NPs	1.0 × 10^−15^ to 1.0 × 10^−9^ M.
[96]	4 DNA bases	Reduced graphene-nanowalls	0.1 fM to 10 mM and 9.4 zM (∼5 dsDNA/mL)
[97]	Leukemia cells	graphene oxide nanoplates and reduced graphene oxide nanowalls	leukaemia fractions (LFs) of ∼10^−11^ LOD
[98]	Leukemia cells	Mg^2+^-charged spongy graphene electrodes	wide concentration range of 1.0 × 10^5^–0.1 cell/mL and LOD of ∼0.02 cell/mL

## Data Availability

Not applicable.

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
