# Peer review of "Smart Graphene-Based Electrochemical Nanobiosensor for Clinical Diagnosis: Review"

_sensors, 2023, doi:10.3390/s23042240_

Round 1
Reviewer 1 Report
A review on potential applications of graphene in electrochemical sensing of diagnoses has been presented. The subject is highly interesting. Therefore, I can recommend its publication. However, there are some points which should be addressed and discussed in the revised version for further completion of the work. Therefore, I suggest revision of the manuscript based on the following points:
1. One of the important application of graphene in electrochemical biosensors is use of graphene nanowalls in label-free detection of single nucleotide polymorphisms of DNA molecules [Toward Single-DNA Electrochemical Biosensing by Graphene Nanowalls], providing early diagnosis of leukemia in blood samples [Ultra-sensitive detection of leukemia by graphene]. These should be mentioned in the revised version of the manuscript for further completion of the review.
2. Table 4 can be completed further by the reports relating to graphene-based electrochemical biosensors. See for example, [Carbon 79 (2014) 654-663] for detection of blood cancer, [Accurate identification of SARS-CoV-2 variant delta using graphene/CRISPR-dCas9 electrochemical biosensor] for detection of SARS-CoV-2, [Graphene oxide for rapid determination of testosterone in the presence of cetyltrimethylammonium bromide in urine and blood plasma of athletes] for detection of testosterone, and [Electrochemical sensor based on nitrogen doped graphene: simultaneous determination of ascorbic acid, dopamine and uric acid] for detection of dopamine and uric acid.
3. I suggest devoting a section relating to the role of graphene in SARS-CoV-2 biosensors, too. In this regard, see for example, [Rapid detection of SARS-CoV-2 using graphene-based IoT integrated advanced electrochemical biosensor], [Electrochemical Biosensor Based on Laser-Induced Graphene for COVID-19 Diagnosing: Rapid and Low-Cost Detection of SARS-CoV-2 Biomarker Antibodies], [ACS Appl. Bio Mater. 2022, 5, 5, 2421–2430] and [Highly stable buffer-based zinc oxide/reduced graphene oxide nanosurface chemistry for rapid immunosensing of SARS-CoV-2 antigens].
4. It has been mentioned that “Mostly, the word grapheme, when mentioned without indicating the form …”. It seems that “grapheme” should be changed to “graphene”.
5. It has been mentioned that “The diverse properties of graphene include flexibility, strength, conductivity (electrically) and its transparency can be employed as a compound for fabrication of torch screen.”. This statement can be supported by a recent article such as [Graphene as the ultra-transparent conductive layer in developing the nanotechnology-based flexible smart touchscreens].
6. It has been mentioned that “in biology, the strength properties of graphene can be of advantage for use as a scaffold for understanding biological materials and molecules [31, 33]”. The phrase “in biology” should be changed to “In biology”. In addition, refs. [31] and [33] seem most relating to the synthesis of graphene rather than their applications as scaffolds. Hence, this statement can be supported by the following articles: [Three-Dimensional Graphene Foams: Synthesis, Properties, Biocompatibility, Biodegradability, and Applications in Tissue Engineering] and [Graphene scaffolds in progressive nanotechnology/stem cell-based tissue engineering of the nervous system].
7. It has been mentioned that “Even with the continuous transformation of science and technology, the industrial scale of the production of graphene is still very expensive, especially, using the exfoliation method.”. But it is not right at all. For example, there are some proposed methods for inexpensive as well as mass-production of graphene (see, for example, [Growth of Graphene from Food, Insects, and Waste] and [Synthesis of graphene from natural and industrial carbonaceous wastes]). This point should be considered in the revised version.
8. Some methods have been mentioned for the synthesis of graphene. However, there are other works such as chemical exfoliation [Synthesis and exfoliation of isocyanate-treated graphene oxide nanoplatelets], thermal shock exfoliation [The effect of heat treatment on formation of graphene thin films from graphene oxide nanosheets], and Li [Ex-MWNTs: Graphene Sheets and Ribbons Produced by Lithium Intercalation and Exfoliation of Carbon Nanotubes] and Na [Graphene Jet Nanomotors in Remote Controllable Self-Propulsion Swimmers in Pure Water] intercalation exfoliation should be added to the manuscript for further completion of the subject.
9. It has been mentioned that “Different types of NPs and NCs are utilized which include liposomes, gold NPs, dendrimers Cdse/ZnS core shells for optical biosensors, paramagnetic and supermagnetic for magnetic biosensors, Gold NPs are integrated on screen printed carbon electrode, single walled carbon nanotubes, Platinum NPs are coated on nonporous film and Au NP based on graphite epoxy composite for electrical biosensors”. This should be improved as “Different types of NPs and NCs are utilized which include liposomes [Pesticide detection with a liposome-based nano-biosensor], gold NPs [Enzymatic Formation of Recombinant Antibody-Conjugated Gold Nanoparticles in the Presence of Citrate Groups and Bacteria], dendrimers CdSe/ZnS core shells for optical biosensors [Anal. Chem. 2011, 83, 10, 3873–3880], paramagnetic and supermagnetic for magnetic biosensors [Magnetite/dextran-functionalized graphene oxide nanosheets for in vivo positive contrast magnetic resonance imaging]. Gold NPs are also integrated on screen printed carbon electrodes [Ultrasensitive cholesterol biosensor based on enzymatic silver deposition on gold nanoparticles modified screen-printed carbon electrode], single walled carbon nanotubes [ACS Appl. Mater. Interfaces 2012, 4, 8, 4312–4319] and graphite epoxy composites [Development of a high analytical performance-tyrosinase biosensor based on a composite graphite–Teflon electrode modified with gold nanoparticles] for designing biosensors.”.
Author Response
Response Report
I would like to thank the editor and the anonymous reviewers for giving me the opportunity to revise and resubmit this manuscript. I carefully studied their comments and incorporated all of them in the revised version. This report shows my point-to-point response per the appreciated reviewers’ comments. BLUE colour shows reviewer’s comment, BLACK AND RED colours show author’s responses
Reviewer 1
Reviewer’s Comments: One of the important applications of graphene in electrochemical biosensors is use of graphene nanowalls in label-free detection of single nucleotide polymorphisms of DNA molecules [Toward Single-DNA Electrochemical Biosensing by Graphene Nanowalls], providing early diagnosis of leukaemia in blood samples [Ultra-sensitive detection of leukemia by graphene]. These should be mentioned in the revised version of the manuscript for further completion of the review.
Response: The 2 articles are added as suggested
The study conducted by Akhavan et al. [107] applied reduced graphene nanowalls for the development of an ultra-high-resolution electrochemical biosensor for detection of the 4 DNA bases as well as monitoring the oxidation signals of the individual nucleotide bases (A, G, C and T). Moreover, the study also compared the electrochemical performances of the modified electrochemical reactivity of the 4 DNA bases, single-stranded DNA, and double-stranded DNA (dsDNA) at the surface of the reduced graphene nanowalls electrode with glassy carbon, graphite and reduced graphene nanosheets electrodes. The electrochemical performances were evaluated on the basis of DPV scans which led to the linear dynamic detection range of 0.1 fM to 10 mM using reduced graphene nanowalls for dsDNA and 2.0 pM to <10 mM using reduced graphene nanosheets respectively. Moreover, the LOD of DsDNA for both reduced graphene nanowalls and reduced graphene nanosheets were estimated as 9.4 zM (∼5 dsDNA/mL) and 5.4 fM, respectively. Comparison between the 2 electrodes in terms of SNP shows that reduced graphene nanowalls were more efficient with 20 zM oligonucleotides (∼10 DNA/mL).
Akhavan et al. [108] developed fast ultrasensitive electrochemical biosensor for the detection of leukaemia cells. The study employed reduced graphene nanowalls electrodes for the detection of leukaemia cells at leukaemia fractions (LFs) of ∼10−11 (as the lower detection limit) in a blood serum containing the extracted guanine. The electrochemical evaluation of the system was conducted using DPV and the signals acquired by oxidation of the extracted guanine on the reduced graphene oxide nanowall electrodes were evaluated after 20 cycles. The result shows that the DPV peaks relating to guanine oxidation (i.e., without guanine extraction) of normal and abnormal cells overlapped at LFs <10−9. However, the DPV result using glassy carbon electrodes was able to detect only LFs ∼ 10−2. The result of this study has claimed that the ultra-sensitivity obtained as a result of the combination of both methods (i.e., guanine extraction by graphene oxide nanoplates and then guanine oxidation by reduced graphene oxide nanowalls) produced higher sensitivity than existing technologies.
Reviewer’s Comments: Table 4 can be completed further by the reports relating to graphene-based electrochemical biosensors. See for example, [Carbon 79 (2014) 654-663] for detection of blood cancer, [Accurate identification of SARS-CoV-2 variant delta using graphene/CRISPR-dCas9 electrochemical biosensor] for detection of SARS-CoV-2, [Graphene oxide for rapid determination of testosterone in the presence of cetyltrimethylammonium bromide in urine and blood plasma of athletes] for detection of testosterone, and [Electrochemical sensor based on nitrogen doped graphene: simultaneous determination of ascorbic acid, dopamine and uric acid] for detection of dopamine and uric acid.
Response: The articles suggested are cited as {112, 113, 114 and 115 respectively}.
The electrochemical detection of leukaemia and normal blood cell using spongy graphene electrode was proposed by Akhavan et al [112]. The biosensor was designed using Mg2+-charged spongy graphene electrodes (SGEs) which was fabricated using electrophoretic deposition of chemically exfoliated graphene oxide sheets on graphite rods. The electrochemical evaluation of the proposed biosensor using DPV resulted in linear dynamic detection behaviour with wide concentration range of 1.0 × 105–0.1 cell/mL and LOD of ∼0.02 cell/mL. Other graphene-based electrochemical biosensors include identification of SARS-COV-2 using dCas9 [113], determination of testosterone [114], determination of ascorbic acid, dopamine and uric acid [115]
Reviewer’s Comments: I suggest devoting a section relating to the role of graphene in SARS-CoV-2 biosensors, too. In this regard, see for example, [Rapid detection of SARS-CoV-2 using graphene-based IoT integrated advanced electrochemical biosensor], [Electrochemical Biosensor Based on Laser-Induced Graphene for COVID-19 Diagnosing: Rapid and Low-Cost Detection of SARS-CoV-2 Biomarker Antibodies], [ACS Appl. Bio Mater. 2022, 5, 5, 2421–2430] and [Highly stable buffer-based zinc oxide/reduced graphene oxide nanosurface chemistry for rapid immunosensing of SARS-CoV-2 antigens].
Response: We are planning to write another review on that topic. However, we highlight some of the contributions of the above articles and cited as [116, 117 and 118].
Rapid detection of SARS-CoV-2 using graphene-based IoT integrated advanced electrochemical biosensor: Sadique et al. [116] provided a review on graphene based IoT integrated electrochemical biosensors for rapid diagnosis of SARS-CoV-2. Despite close similarities with the current review, the study does not cover broad topics and open research issue.
Reviewer’s Comments: It has been mentioned that “Mostly, the word grapheme, when mentioned without indicating the form …”. It seems that “grapheme” should be changed to “graphene”.
Response: Thanks a lot for this observation, the word “grapheme” is changed to “graphene”.
Reviewer’s Comments: It has been mentioned that “The diverse properties of graphene include flexibility, strength, conductivity (electrically) and its transparency can be employed as a compound for fabrication of torch screen.”. This statement can be supported by a recent article such as [Graphene as the ultra-transparent conductive layer in developing the nanotechnology-based flexible smart touchscreens].
Response: The statement is supported as suggested {see Ref 109}.
Reviewer’s Comments: It has been mentioned that “in biology, the strength properties of graphene can be of advantage for use as a scaffold for understanding biological materials and molecules [31, 33]”. The phrase “in biology” should be changed to “In biology”. In addition, refs. [31] and [33] seem most relating to the synthesis of graphene rather than their applications as scaffolds. Hence, this statement can be supported by the following articles: [Three-Dimensional Graphene Foams: Synthesis, Properties, Biocompatibility, Biodegradability, and Applications in Tissue Engineering] and [Graphene scaffolds in progressive nanotechnology/stem cell-based tissue engineering of the nervous system].
Response: Thanks for the observation, in biology is changed to In biology as suggested. Subsequently, the statement is supported as suggested {see Ref 110 and 111}.
Reviewer’s Comments: It has been mentioned that “Even with the continuous transformation of science and technology, the industrial scale of the production of graphene is still very expensive, especially, using the exfoliation method.”. But it is not right at all. For example, there are some proposed methods for inexpensive as well as mass-production of graphene (see, for example, [Growth of Graphene from Food, Insects, and Waste] and [Synthesis of graphene from natural and industrial carbonaceous wastes]). This point should be considered in the revised version.
Response: Thanks for this vital comment. The paragraph is adjusted as suggested
Over the years, scientists have developed several approaches for the production of graphene. These methods range from exfoliation method which is expensive to less expensive approaches such as the production of graphene from food, insects, waste as well as and synthesis of graphene from natural and industrial carbonaceous wastes.
Reviewer’s Comments: Some methods have been mentioned for the synthesis of graphene. However, there are other works such as chemical exfoliation [Synthesis and exfoliation of isocyanate-treated graphene oxide nanoplatelets], thermal shock exfoliation [The effect of heat treatment on formation of graphene thin films from graphene oxide nanosheets], and Li [Ex-MWNTs: Graphene Sheets and Ribbons Produced by Lithium Intercalation and Exfoliation of Carbon Nanotubes] and Na [Graphene Jet Nanomotors in Remote Controllable Self-Propulsion Swimmers in Pure Water] intercalation exfoliation should be added to the manuscript for further completion of the subject.
Response: The method suggested are highlighted in the paragraph below:
The study proposed by Stankovich et al. [120] revolves around the treatment of GO with Organic Isocyanates which can be exfoliated into functionalized graphene oxide nanoplatelets. Akhavan [121] proposed the production of Graphene thin films with very low concentration of oxygen-containing functional groups. These graphene thin films are produced via the reduction of GO nanosheets (i.e., produced via chemical exfoliation) in a reducing medium as well as one and two-step heat treatment methods. Cornor- Márquez et al. [122] proposed the production of graphene flakes, multi-layered flat graphitic structures (nanoribbons), and partially opened multi-walled carbon nanotubes via the intercalation of lithium and ammonia followed by exfoliation of multi-walled carbon nanotubes. While the study proposed by Akhavan [123] for the first time demonstrated the synthesis of remote controllable working graphite nanostructured swimmer based on graphite het nanomotor.
Reviewer’s Comments: It has been mentioned that “Different types of NPs and NCs are utilized which include liposomes, gold NPs, dendrimers Cdse/ZnS core shells for optical biosensors, paramagnetic and supermagnetic for magnetic biosensors, Gold NPs are integrated on screen printed carbon electrode, single walled carbon nanotubes, Platinum NPs are coated on nonporous film and Au NP based on graphite epoxy composite for electrical biosensors”. This should be improved as “Different types of NPs and NCs are utilized which include liposomes [Pesticide detection with a liposome-based nano-biosensor], gold NPs [Enzymatic Formation of Recombinant Antibody-Conjugated Gold Nanoparticles in the Presence of Citrate Groups and Bacteria], dendrimers CdSe/ZnS core shells for optical biosensors [Anal. Chem. 2011, 83, 10, 3873–3880], paramagnetic and supermagnetic for magnetic biosensors [Magnetite/dextran-functionalized graphene oxide nanosheets for in vivo positive contrast magnetic resonance imaging]. Gold NPs are also integrated on screen printed carbon electrodes [Ultrasensitive cholesterol biosensor based on enzymatic silver deposition on gold nanoparticles modified screen-printed carbon electrode], single walled carbon nanotubes [ACS Appl. Mater. Interfaces 2012, 4, 8, 4312–4319] and graphite epoxy composites [Development of a high analytical performance-tyrosinase biosensor based on a composite graphite–Teflon electrode modified with gold nanoparticles] for designing biosensors.”.
Response: The paragraph is adjusted as suggested
Different types of NPs and NCs are utilized which include liposomes [124], gold NPs [125], dendrimers CdSe/ZnS core shells for optical biosensors [126], paramagnetic and supermagnetic for magnetic biosensors [127]. Gold NPs are also integrated on screen printed carbon electrodes [128], single walled carbon nanotubes [129] and graphite epoxy composites [130] for designing biosensors

Reviewer 2 Report
Journal Name: Sensors-MDPI
Title: Smart Graphene-Based Electrochemical Nanobiosensor For Clinical Diagnosis: Review
In the current review, Y. W. Hartati et al., have reviewed the recent advances in the smart graphene-based electrochemical nano biosensor for clinical diagnosis.
This review is useful and the theme of the review matches the scope of MDPI sensors. However, much of the contents in this manuscript is just a general description, a detailed personal point of view is not given. Based on the above considerations, a major revision is given. Here are some suggestions.
Major revision
1. The graphical abstract is necessary
2. Kindly provide the research highlights
3. I feel authors can give an abbreviation table and contents section
4. Abstract can be further improved
5. The authors have to make a small subsection and discuss the advantages of electrochemical sensors over other sensor technologies.
6. Discuss the different forms of carbon and how each type of carbon allotropes has different types of effects as an electrochemical transducer.
7. The authors have to discuss a few recent articles related to carbon in sensing applications (A Short Review Comparing Carbon‐Based Electrochemical Platforms With Other Materials For Biosensing SARS‐Cov‐2)
8. Also discuss how theoretical chemistry helps develop electrochemical sensor technologies.
9. The conclusion is too simple it can be improved further
Author Response
Response Report
I would like to thank the editor and the anonymous reviewers for giving me the opportunity to revise and resubmit this manuscript. I carefully studied their comments and incorporated all of them in the revised version. This report shows my point-to-point response per the appreciated reviewers’ comments. BLUE colour shows reviewer’s comment, BLACK AND RED colours show author’s responses
Reviewer 2
Reviewer’s Comments: The graphical abstract is necessary
Response: The graphical abstracts is shown below
Reviewer’s Comments: Kindly provide the research highlights
Response: The research highlights are outlined below:
Highlights:
- Point-of-care diagnosis is crucial for management of infectious diseases.
- Integration of nanotechnology into biosensing technology increases conductivity, sensitivity and Limit of Detection (LOD).
- Graphene-based electrochemical biosensors have emerged as one of the best approaches for enhancing biosensing technology.
- Integration of Internet of Medical Things (IoMT) in the development of biosensors have the potential to improve detection of diseases and treatments.
Reviewer’s Comments: I feel authors can give an abbreviation table and contents section
Response: The abbreviations are given after conclusion and acknowledgments
Reviewer’s Comments: Abstract can be further improved
Response: We overview the abstract again. We feel like it captures all the concepts limited in this review.
Reviewer’s Comments: The authors have to make a small subsection and discuss the advantages of electrochemical sensors over other sensor technologies.
Response: The advantage of Electrochemical biosensors is discussed as follows:
2.5 Advantages of Electrochemical Biosensors
The application electrochemical biosensors continue to grow due to their prospects in clinical diagnosis, environmental monitoring and food processing. The field of electro-chemical-based biosensors are growing exponentially due to their advantages over other types of biosensor as well as the integration of nanomaterials such as gold NPs, graphene and its derivatives, quantum dots, fullerenes and other carbon allotropes. Some of the inherent advantages of electrochemical biosensors include their excellent detection limits, easy miniaturization easy miniaturization (ease of fabrication), robustness, their ability to be utilized in biofluid with optically absorbing and fluorescing features, evaluation using convenient drawn samples also known as liquid biopsies while simultaneously solving most of the limitations of contemporary approaches to accomplish high level of perfor-mance ( such as faster detection, sensing low amount or concentration of target. As re-ported by Anik et al. [132] Electrochemical biosensors offers high sensitivity, practicality and fast response. These attributes make them suitable as lab-on chips for point of care detection. Electrochemical biosensors are currently in use for detection of wide array of biomolecules present in food samples and human body such as glucose, DNA, lactate, hemoglobin, alanine aminotransferase, uric acid, blood ketones, acetylcholine, cholesterol, cardiac troponin etc.
Reviewer’s Comments: Discuss the different forms of carbon and how each type of carbon allotropes has different types of effects as an electrochemical transducer.
Response: The concept suggested is discussed below
Carbon-allotropes
The last few years have witnessed the Increase application of Carbon-allotropes in biomedical sensing. Carbon-allotropes comes in different varieties which include diamonds, fullerenes, lonsdaleite, graphite, different forms of graphene, nanotubes and nanohorns etc. carbon-allotrope nanomaterials are currently use in several applications due to their unique properties. Some of these applications include biosensing, drug delivery, tissue engineering, bioimaging, medical diagnostic, cancer therapy etc. [133].
In the field of biosensing technology, Carbon-allotropes are used extensively due to their inimitable properties such as electrical, optical and mechanical properties as well as their flexibility, thermal stability, high electron mobilities, strength-to-weight ratio. These properties make them suitable for miniaturizing sensors with low power drainage and high performance. Moreover, apart from their high sensitivity, carbon-allotrope based biosensors offer higher special resolution in regards to localized detection along with POC, non-destructive and label free sensing. Some of the recent carbon allotropes reported in the field of biosensors include carbon nanotubes, fullerene, quantum dots, graphene and its derivatives (such as reduced graphene and GO), nanodiamonds etc. [133-134].
Reviewer’s Comments: The authors have to discuss a few recent articles related to carbon in sensing applications (A Short Review Comparing Carbon‐Based Electrochemical Platforms with Other Materials for Biosensing SARS‐Cov‐2).
Response: The study suggested is added among the comparison with similar studies
The review provided by Soni et al. [119] is limited to the application of Carbon-Based electrochemical biosensor for the detection of SARS-Cov-2. The review covers several aspects such as graphene, electrochemical and nanobiosensors and AI-based detection. However, some of the dissimilarities with this review is the limitation on COVID-19, the absence of IoMT-based platforms and open research issues.
Reviewer’s Comments: Also discuss how theoretical chemistry helps develop electrochemical sensor technologies.
Response: The paragraph below discusses about the role of theoretical chemistry in electrochemical biosensors
Understanding of the basic principle behind atoms, elements, compounds and their reactions allow scientists to develop several applications ranging from diverse chemical products, medical procedures, fuels, batteries etc. The field of theoretical chemistry con-tinue to unravel mysteries surrounding chemical compounds, their constituents, prop-erties, reaction etc. Over the years scientists have conducted several experiments sup-ported by theoretical chemistry which led to the development of several principles, products, and inventions. The field of electrochemistry was revolutionized by Faraday’s two laws which include (1) The amount of a substance deposited on each electrode of an electrolytic cell is directly proportional to the quantity of electricity passed through the cell. (2) The quantities of different elements deposited by a given amount of electricity are in the ratio of their chemical equivalent weights [21, 22, 135].
Reviewer’s Comments: The conclusion is too simple it can be improved further
Response: The conclusion is structured in that manner because we covered extensive issues in “open research issues”.

Reviewer 3 Report
The results of this comprehensive analysis will be helpful for understanding the crucial role of graphene-based electrochemical biosensors coupled with Artifical Intelligence AI and IoMT for clinical diagnostics. It should undergo revision before this work can be published in a scientific journal.
1. There are only 2-3 references of Year 2022 are considered out of 106 references in the proposed review work. Authors should update the review work with recent literatures.
2. There are several literatures based on graphene-based electrochemical biosensors. What is novelty in this work?
3. Authors should prepare a graph to showcase the progress of graphene-based electrochemical biosensors.
4. Authors should also add one good schematic to showcase the complete review work.
5. Authors have discussed the several work separately. Authors should discuss those merits and limitations and also connect the paragraphs with each other.
6. Authors should also discuss the recent references of biosensors especially based on gold nanoparticles that should be relevant to proposed work for the detection of several similar biomolecules, including creatinine, cardiac troponin I, alanine aminotransferase, acetylcholine, cholesterol, uric acid, and p-cresol using localized surface plasmon resonance (LSPR) principle.
7. Authors should add one table and compare the several biosensors.
8. Authors should add some more high-quality relevant images also mention the copyright statements with captions of figures.
Author Response
Response Report
I would like to thank the editor and the anonymous reviewers for giving me the opportunity to revise and resubmit this manuscript. I carefully studied their comments and incorporated all of them in the revised version. This report shows my point-to-point response per the appreciated reviewers’ comments. BLUE colour shows reviewer’s comment, BLACK AND RED colours show author’s responses
Reviewer 3
Reviewer’s Comments: " In There are only 2-3 references of Year 2022 are considered out of 106 references in the proposed review work. Authors should update the review work with recent literatures.
Response: The references were updated (i.e., from 100 to 134).
Reviewer’s Comments: There are several literatures based on graphene-based electrochemical biosensors. What is novelty in this work?
Response: The novelty of this study is already established in the comparison with similar studies where we explained how existing studies differs with the current one. However, novelty of this work can be seen in the study’s highlights
Highlights:
- Point-of-care diagnosis is crucial for management of infectious diseases.
- Integration of nanotechnology into biosensing technology increases conductivity, sensitivity and Limit of Detection (LOD).
- Graphene-based electrochemical biosensors have emerged as one of the best approaches for enhancing biosensing technology.
- Integration of Internet of Medical Things (IoMT) in the development of biosensors have the potential to improve detection of diseases and treatments.
Reviewer’s Comments: Authors should prepare a graph to showcase the progress of graphene-based electrochemical biosensors.
Response: We couldn’t find relevant information regarding the abovementioned concept. The timeline of biosensors and graphene can’t be fit into single graph as highlighted in the figures below. However, table 4 present several electrochemical graphene-based biosensors.
Reference |
Sample/Analyte/Target |
Type of Graphene |
LOD/LR |
[68] |
Lead in blood sample |
gFET |
LOD below 37.5ng/L |
[69] |
Cancer |
gFET |
LOD of less than 100pg/ML |
[70] |
Human papillomavirus |
graphene-polyaniline |
2.3 nM LOD and 10-200 nM LR |
[71] |
Cardiac troponin |
rGO, silver nanoparticles and Molybdenum sulfide (MOS) |
0.3pg/ml to 0.2ng/ml LR |
[72] |
Breast cancer biomarkers |
GO and gold nanoparticles |
0.16nM and 0.23nM LOD and 378nA/nM and 219nA/nM for ERBB2 and CD24 respectively |
[73] |
Living cell peroxide |
RGO-Pt |
0.2Um LOD and 0.5uM to3.475mM wide LR |
[74] |
Mycobacterium tuberculosis |
RGO- gold NPs |
1.0 X 10-15 to 1.0 X 10-9M. |
[107] |
4 DNA bases |
Reduced graphene-nanowalls |
0.1 fM to 10 mM and 9.4 zM (∼5 dsDNA/mL) |
[108] |
Leukemia cells |
graphene oxide nanoplates and reduced graphene oxide nanowalls |
leukaemia fractions (LFs) of ∼10−11 LOD |
[112] |
Leukemia cells |
Mg2+-charged spongy graphene electrodes |
wide concentration range of 1.0 × 105–0.1 cell/mL and LOD of ∼0.02 cell/mL |
Reviewer’s Comments: Authors should also add one good schematic to showcase the complete review work.
Response: The schematic is provided as graphical abstract as shown below:
Reviewer’s Comments: Authors have discussed the several work separately. Authors should discuss those merits and limitations and also connect the paragraphs with each other.
Response: Section 5 title “Hyphenation” connect all the dots ranging from Electrochemical Biosensors and Nanomaterials, Graphene-based biosensors to Blending IoT and AI with Medical Devices. These concepts form the basis of this article.
Reviewer’s Comments: Authors should also discuss the recent references of biosensors especially based on gold nanoparticles that should be relevant to proposed work for the detection of several similar biomolecules, including creatinine, cardiac troponin I, alanine aminotransferase, acetylcholine, cholesterol, uric acid, and p-cresol using localized surface plasmon resonance (LSPR) principle.
Response: The concepts suggested is highlighted under advantages of electrochemical biosensors:
The application electrochemical biosensors continue to grow due to their prospects in clinical diagnosis, environmental monitoring and food processing. The field of electro-chemical-based biosensors are growing exponentially due to their advantages over other types of biosensor as well as the integration of nanomaterials such as gold NPs, graphene and its derivatives, quantum dots, fullerenes and other carbon allotropes. Some of the inherent advantages of electrochemical biosensors include their excellent detection limits, easy miniaturization easy miniaturization (ease of fabrication), robustness, their ability to be utilized in biofluid with optically absorbing and fluorescing features, evaluation using convenient drawn samples also known as liquid biopsies while simultaneously solving most of the limitations of contemporary approaches to accomplish high level of performance ( such as faster detection, sensing low amount or concentration of target. As re-ported by Anik et al. [132] Electrochemical biosensors offers high sensitivity, practicality and fast response. These attributes make them suitable as lab-on chips for point of care detection. Electrochemical biosensors are currently in use for detection of wide array of biomolecules present in food samples and human body such as glucose, DNA, lactate, hemoglobin, alanine aminotransferase, uric acid, blood ketones, acetylcholine, cholesterol, cardiac troponin etc.
Reviewer’s Comments: Authors should add one table and compare the several biosensors.
Response: The comparison of different types of biosensors are presented in table 2.
Biosensor |
Types |
Advantages |
Disadvantages |
Electrochemical |
Amperometric, potentiometric, conductimetric, impedimetric and voltametric |
excellent detection limits, easy miniaturization easy miniaturization (ease of fabrication), robustness, faster detection, linear output, good resolution etc. |
Unstable current and voltage, narrow or limited temperature range, cross sensitivity with other gases, short or limited shelf life |
Optical |
surface plasmon resonance (SPR), fluorescence, bio/chemiluminescence, refractive index, Raman scattering, absorbance |
high sensitivity, selectivity, cost-effectiveness, small size |
Susceptible to physical change and interference from environmental effects |
Thermal/calorimetric |
Thermistors or thermopiles |
Scalability, Ease of use and ease of fabrication |
Lack of specificity inn temperature measurements, long experimental procedures |
Mass sensitive or gravimetric |
Wave biosensor, surface acoustic, cantilever |
Low cost and simplicity |
Low sensitivity |
Piezoelectric |
Surface acoustic devices, Piezoelectric crystal |
Fast detection, good frequency response, small size and high sensitivity |
High temperature sensitivity, not suitable for static conditions, some crystal can dissolve in high humid environment and are water soluble |
Reviewer’s Comments: Authors should add some more high-quality relevant images also mention the copyright statements with captions of figures.
Response: The issue of copyright is very critical and concern for the journal, thus, we prepare to draw our own figures because seeking permission to use existing figures is very challenging and may take time.

Round 2
Reviewer 1 Report
The manuscript has been revised based on the comments and now it is publishable.
Reviewer 2 Report
The authors made sufficient changes article can be accepted in the current form.
Reviewer 3 Report
Satisfactory revision.